# Savoring the present: The reciprocal influence between positive emotions and positive emotion regulation in everyday life

**Desirée Colombo**[1]*, **Jean-Baptiste Pavani**[2], **Javier Fernandez-Alvarez**[3], **Azucena Garcia-Palacios**[1,4], **Cristina Botella**[1,4]

**1** Department of Basic Psychology, Clinic and Psychobiology, Jaume I University, Castellón, Spain, **2** Center for Research on the Psychology of Cognition, Language and Emotion (PSYCLE), Aix Marseille University, Marseille, France, **3** Department of Psychology, Catholic University of the Sacred Heart, Milan, Italy, **4** CIBER Fisiopatología Obesidad y Nutrición (CIBERobn), Instituto Salud Carlos III, Madrid, Spain

* dcolombo@uji.es

**Data Availability Statement:** The data supporting the findings of this study are openly available at https://doi.org/10.17605/OSF.IO/TEUBR.

## Abstract

A growing body of research has investigated the regulation of negative emotions in ecological settings, but little is known about the mechanisms underlying positive emotion regulation in everyday life. Although some evidence suggests that adopting positive strategies is beneficial for emotional well-being, the literature is inconsistent about the effects of positive emotions on subsequent regulatory processes. In the present study, we adopted a two-week ecological momentary assessment to explore the association between positive emotions and positive emotion regulation in daily life. According to our results, the less individuals felt positive emotions at one point, the more they tended to enhance their use of positive strategies from this time to the next, which in turn resulted in subsequent higher levels of positive emotions. This prototype of positive regulation can be seen as a highly adaptive mechanism that makes it possible to compensate for a lack of positive emotions by enhancing the deployment of positive strategies. The theoretical and clinical implications of these findings are discussed.

## 1. Introduction

The pursuit of happiness is considered one of the most important life goals of individuals [1], who intensely seek to create pleasant experiences throughout their lives. Positive emotions (PE) are a core component of well-being because they are not limited to pleasant sensations, but rather produce short- and long-term psychological benefits and improve both physical and mental health [2–5]. More specifically, PE temporarily extend the scope of attention, cognition and action [6], which in turn promotes resilience and psychological well-being [7]. Accordingly, people spend most of their time trying to downregulate negative emotions and upregulate positive ones [8].

Emotion regulation is a process through which individuals try to influence their emotional state in order to achieve personal goals [9]. To date, most of the literature has focused on the

**Funding:** This work was supported by the Marie Curie EF-ST AffecTech Project, approved at call H2020 – MSCA – ITN – 2016, under grant number 722022.

**Competing interests:** The authors have declared that no competing interests exist.

regulation of negative emotional states. Nevertheless, there is increasing evidence highlighting the crucial role of positive emotion regulation [10,11], that is, the set of strategies people implement to create, maintain and enhance PE for two main purposes. First, people upregulate PE for its own sake, that is, to experience pleasurable states and increase happiness [12]. Second, the upregulation of PE has been recognised as a mood repair mechanism, i.e., a process that helps individuals to reduce negative affect and recover from stressful events [13,14].

Although originally conceptualized for negative emotion, Gross' extended model (2015) [15] has also been used to understand positive emotion regulation [10]. Accordingly, different types of positive strategies can be deployed in different stages of the emotion generation process: (a) by selecting a situation that is expected to improve affect (situation selection); (b) by actively changing a situation in order to get the most out of it (situation modification); (c) by redirecting the attention towards specific features or details of a situation that might increase positive emotions (attentional deployment); (d) by changing the appraisal of an emotion-eliciting stimulus in order to amplify the associated pleasant state (cognitive change); and (e) by experientially, physiologically, or behaviourally expressing ongoing PE to further increase their intensity (response modulation). These strategies are implemented not only during the experience of a positive emotional state [10]. They might also be used before (i.e., while anticipating a positive event) [16] or after (i.e., while recalling a positive memory) [17] the emotion-generative process [10,11]. For the purposes of the present study, however, we will mainly focus on the available literature exploring PE regulation in the present.

When assessed in naturalistic settings, people use on average a repertoire of sixteen strategies in response to PE [18]. The implementation of positive emotion regulation has been shown to be beneficial for mental health, and a growing body of studies has revealed that people who frequently adopt strategies to intensify and prolong positive experiences (that is, savoring [19]) show enhanced emotional well-being [20,21] and more sustained PE over time [22]. For instance, a more extensive use of some strategies, such as counting blessings or sharing, leads to greater levels of happiness, despite experiencing few daily positive events [21,23]. In another study, Langston et al. [24] found that capitalizing on positive events (i.e., the process of beneficially seizing and interpreting positive situations) further increases the experience of positive emotions. Furthermore, the intense use of strategies, such as mindfulness and positive reappraisal, has been found to predict higher levels of psychological well-being and enhanced experience of positive emotional states [25,26]. In sum, there is a growing body of evidence that highlights the important emotional outcomes associated with the use of positive emotion regulation in daily life.

However, not only can emotion regulation influence emotional outcomes, but emotions can also determine subsequent emotion regulation processes [27]. This hypothesis is further confirmed by the evidence showing that momentary mood predicts subsequent affect levels [28], which suggests that emotion regulation might be partly determined by an individual's momentary emotional state [29]. Nonetheless, the previous literature has been inconsistent about the association between PE and positive emotion regulation.

On the one hand, the broaden-and-build theory states that the experience of PE enhances one's attentional scope and thought-action repertoire, leading to cognitive and behavioural broadening mechanisms [7,30] such as increased creativity or cognitive flexibility [31,32]. These mechanisms have been hypothesized to affect emotion regulation processes as well. Thus, positive emotions are likely to encourage the adoption of adaptive, broadminded strategies that further enhance positive states [7,33]. Consistent with this theory, the momentary experience of high levels of PE has been found to predict greater subsequent adoption of adaptive strategies such as problem solving [29] and mindfulness [34].

On the other hand, pro-hedonic theories, such as the hedonic flexibility principle, suggest that people are likely to implement behavioural strategies based on their momentary mood in

order to minimize negative affect and maximize positive emotions [35,36]. More specifically, the experience of low positive emotions is supposed to motivate actions and behaviours to enhance mood. Thus, individuals are likely to increase their attempts to upregulate PE when experiencing a low rather than high level of PE. Recent studies have demonstrated that, when experiencing bad moods, people are more likely to engage in mood-enhancing activities, such as doing sport, going out in nature or chatting with a friend, whereas useful but mood-decreasing activities are pursued when the current mood is already high [36,37]. In another study, individuals were found to seek pleasant social relationships when feeling bad and prefer solitude or less pleasant social interactions when feeling good [38].

In sum, despite the growing evidence highlighting the importance of PE in mental health, there are still many unanswered questions about the regulatory mechanisms underlying positive states. Although positive emotion regulation has received increasing attention in the past decade, the effects of its momentary use are still unexplored. More specifically, whereas the findings about the emotional outcomes of positive emotion regulation are quite consistent, the effect of momentary PE on subsequent strategy implementation is still largely unknown. Importantly, momentary PE not only reflect the experience of a pleasant state, but they also represent an important source of information that drives regulatory mechanisms. Thus, exploring the reciprocal influences between PE and positive emotion regulation is important in order to disentangle the factors determining past, present and future positive emotional experiences.

## The current study

The aim of the current study was to explore the reciprocal interconnection between PE and positive emotion regulation in daily life. To do so, we asked 85 undergraduate students to use a two-week Ecological Momentary Assessment (EMA) to report their momentary levels of PE and rate the adoption of positive strategies to regulate ongoing PE. Despite the evidence showing that people's repertoire of positive strategies is quite large [18], it would have been too demanding and time-consuming to assess a high quantity of items at each assessment. Accordingly, we decided to focus on a limited number of positive strategies and to exclude dampening ones (i.e., strategies that decrease the intensity of ongoing PE).

The rational adopted for selecting the strategies was based on Quoidbach et al.'s theory (2015) of positive emotion regulation. While the effectiveness of situation selection and situation modification strategies to enhance PE has been found to be weak or even largely unknown and controversial, there is more consistent evidence that supports the value of attentional deployment, cognitive change and response modulation strategies to increase positive emotions, especially in the short-term [10]. We therefore decided to focus on these three categories and, for each of them, two strategies were selected based on the previous literature relating positive emotion regulation to PE, thus making a total of 6 strategies: mindfulness, stimulus control, broadening, counting blessings, emotion expression, and sharing. The use of this rational allowed us to explore the association between PE and positive emotion regulation both at a strategy and category levels.

Attentional deployment refers to the set of strategies specifically designed to direct one's attention in order to savour a pleasant emotional state, which in turn increases the experience of positive states both in the short and long terms [10]. In the present study, we explored the momentary use of mindfulness, which is focusing the attention on the present situation, and stimulus control, which refers to the attempt to avoid other negative thoughts in order to focus on the pleasant state. According to the previous literature, both strategies can play a role in daily positive emotional states. Indeed, a growing body of literature has found mindfulness to be associated with more intense and frequent positive emotions [20,25,39], which in turn has been shown to increase next-day mindfulness levels [34]. Furthermore, Heiy and Cheavens

[18] found stimulus control to be one of the most frequently adopted strategies in response to PE that positively affect general mood.

Cognitive change refers to the attempt to influence the meaning of a positive stimulus, for example, by reappraising a positive situation as a special moment or by increasing the value attributed to a positive event. Generally, previous research has shown that reappraising a positive stimulus [40,41] and increasing the perceived value of a positive experience [42] are associated with enhanced levels of positive emotions. Indeed, not only does the adoption of cognitive change strategies enhance momentary positive emotional states, but it also has a modest impact on PE in the long-terms [10]. In the present study, we assessed participants' use of broadening, which is thinking about the current pleasant state as part of a worthy life, and counting one's blessings, i.e., thinking about the special moments by not taking them for granted. Whereas one's perceived satisfaction and fulfilment in different aspects of life have been found to significantly affect emotional well-being [43], Wood et al. (2010) [44] showed the significant effect of counting one's blessings on increasing positive emotions.

Finally, response modulation includes strategies to influence the physiological, experiential or behavioural response to a positive state, which usually involve expressing the emotion with either verbal or nonverbal communication. Indeed, the accumulated literature coming from embodied cognition research has suggested that expressing positive emotions both physically– for example, by facial display [45,46])—and verbally [24,47] can boost the experience of the associated positive state. Consistently, we assessed the momentary use of emotion expression (i.e., the use of the body to express and communicate ongoing PE) as well as sharing (i.e., the tendency to share positive experiences through verbal communication with other people).

The first objective of this study was to explore which of the two aforementioned theories better explains the association between PE and positive emotion regulation strategies. According to the broaden-and-build theory, the experience of intense PE fosters broadening mechanisms and the use of broad-minded positive strategies. In this case, and consistent with previous findings, high levels of PE should determine an increase in the subsequent use of positive strategies. In contrast, the hedonic flexible principle states that low mood, compared to high mood, predicts the implementation of strategies to enhance momentary emotions. Thus, a lower level of PE at a certain time should predict an increase in the use of positive emotion regulation from that time to the next.

The second objective of this study was to explore the unique impact of six positive emotion regulation strategies on subsequent PE. Consistent with the ample evidence showing the beneficial emotional outcomes of positive regulation, we expected to find that increased use of positive strategies at one point predicted enhanced PE in the following assessment.

The third objective was to investigate whether the reciprocal influence between PE and positive emotion regulation changes significantly depending on the intrinsic nature of strategies, as defined by the three categories explored in the present study: attentional deployment, cognitive change and response modulation. To this aim, we explored whether strategy category significantly moderated the association between positive emotion regulation and PE and, more specifically: (a) whether the strategy category moderated the impact of PE at t0 on positive regulation at t1, and (b) whether the strategy category moderated the effect of positive regulation on subsequent levels of PE.

## 2. Material and methods

### 2.1 Inclusion criteria and sample

In order to exclude the potential confounding effect of depression, which has been shown to be associated with an impaired use of savoring strategies and an increased adoption of

dampening strategies [48,49], individuals with a score above 14 on the Patient Health Questionnaire (PHQ-9) [50] were excluded from the study (i.e., individuals with moderate to severe depressive conditions; *n* = 6). Similar to the sample size of previous EMA studies on emotion regulation [29,51], we recruited 85 undergraduate students at Jaume I University (Castellon, Spain). The sample was composed of 67 females (77.9%) and 19 males (22.1%); their ages ranged between 18 and 36 years (*M*: 22.07; *SD*:3.45).

This study was approved by the ethics committee of Jaume I University (certificate number: CD/57/2019), and informed consent was obtained from each participant.

## 2.2 Material

Participants were prompted three times a day for two weeks to complete a brief questionnaire on their smartphone, reaching a total of 42 potential observations for each participant. Consistent with previous studies, this sampling frequency has been shown to be adequate for the assessment of daily emotion regulation patterns [18] and it leads to good compliance levels [16,17]. In the present study, 2726 out of 3570 possible assessments were obtained, thus revealing a mean compliance of 76.34% (*SD* = 18.12), ranging between 33% and 100%.

At each prompt, participants were first asked to rate the momentary intensity of seven PE on a Likert scale from 1 to 5 (1 = *not at all*, 5 = *a lot*). The seven emotions were selected in order to include both low-arousal (hope, serenity, gratitude) and high-arousal (happiness, amusement, excitement, pride) positive emotions [52], which is consistent with the evidence showing the influence of positive regulation on both types of positive emotions [26]. To obtain a general indicator of PE, the seven positive emotions rated at each assessment were averaged. The composite score obtained showed high internal consistency at both the between- ($\alpha$ = .96) and within-individual levels ($\alpha$ = .86).

Participants were also asked to rate the momentary adoption of six positive strategies on a 0–100 scale (0 = *no adoption*, 100 = *high adoption*), which were selected with the aim of exploring three categories of positive regulation [10]: mindfulness and stimulus control for the category 'attentional deployment'; broadening and counting blessings for the category 'cognitive change'; emotion expression and sharing for the category 'response modulation'. Due to the lack of validated questionnaires to assess positive strategies, ad hoc single items were created (see **Table 1**), as is common in ecological studies exploring emotion regulation [see, for example, 25,30,32,38]. These items were mostly inspired by a previous study [18].

Similar to previous EMA studies exploring the reciprocal influences between emotion and emotion regulation [see for example 28,35,43], change scores were calculated for each strategy, indicating whether a strategy was used more or less at a certain time (t1), compared to the previous assessment (t0). These scores were calculated to analyse to what extent PE at t0 influenced positive emotion regulation change from t0 to t1. To compute change scores that are not affected by the so-called 'regression toward the mean effect', change scores were computed through linear mixed-effects models with maximum likelihood by taking the residuals of a model in which the strategy at t1 was regressed on itself at t0. In addition, strategy type was also taken into account in the analyses to explore further the relationship between PE and positive regulation depending on the intrinsic nature of the strategies adopted. To do so, strategies were averaged based on their category. This made it possible to obtain three new variables that reflected the intensity of use of each category of strategy. To assess the internal consistency of the new variables, correlations for each pair of strategies were performed at the between-individual level (attentional deployment: *r* = .882, *p* < .001; cognitive change: *r* = .981, *p* < .001; response modulation: *r* = .936, *p* < .001) and within-individual level (attentional deployment: *r* = .556, *p* < .001; cognitive change: *r* = .732, *p* < .001; response modulation: *r* = .630, *p* < .001).

**Table 1. Correlations between emotion regulation and PE at the within-individual level.**

| | M (SD) | 1 | 2 | 3 | 4 | 5 | 6 | 7 | 8 | 9 |
|---|---|---|---|---|---|---|---|---|---|---|
| 1. PE | 2.77 (0.97) | 1.00 | | | | | | | | |
| STRATEGIES | | | | | | | | | | |
| 2. Mindfulness: *I'm trying to be focused on the present and concentrate on how good I feel* | 56.34 (27.06) | .538*** | 1.00 | | | | | | | |
| 3. Stimulus control: *I'm trying to avoid all negative thoughts and stressors in order to focus and make the most of my positive emotions* | 52.5 (28.89) | .400*** | .556*** | 1.00 | | | | | | |
| 4. Broadening: *I'm thinking about all the good things I have and that are happening in my life as well* | 51.65 (28.96) | .489*** | .601*** | .565*** | 1.00 | | | | | |
| 5. Counting blessings: *I'm thinking about how lucky I am to live in this moment and feel so good* | 51.35 (29.02) | .504*** | .625*** | .558*** | .732*** | 1.00 | | | | |
| 6. Emotion expression: *I'm trying to express and emphasize my emotions on the outside by showing them* | 46.95 (30.39) | .422*** | .462*** | .368*** | .474*** | .463*** | 1.00 | | | |
| 7. Sharing: *I'm sharing my positive emotions with other people, for example, with my friends, partner, and/or family* | 44.59 (31.98) | .406*** | .422*** | .377*** | .468*** | .452*** | .630*** | 1.00 | | |
| CATEGORIES | | | | | | | | | | |
| 8. Attentional deployment | 54.43 (25.93) | .520*** | .854*** | .891*** | .646*** | .650*** | .454*** | .444*** | 1.00 | |
| 9. Cognitive change | 51.51 (28.03) | .527*** | .653*** | .591*** | .923*** | .931*** | .497*** | .489*** | .686*** | 1.00 |
| 10. Response modulation | 45.81 (29.57) | .454*** | .487*** | .409*** | .514*** | .500*** | .896*** | .899*** | .494*** | .539*** |

*$p < .05$,

** $p < .01$,

*** $p < .001$.

Means and standard deviations were computed on raw variables. Categories were obtained by averaging strategies in the following way: mindfulness and stimulus control for 'attentional deployment', broadening and counting blessings for 'cognitive change', and emotion expression and sharing for 'response modulation'. (PE: Positive emotions).

## 2.3 Procedure

Participants were recruited via social media and poster advertisements placed in different buildings at the university. Students willing to participate were invited to visit the laboratory to receive more details about the study design and sign the informed consent.

The EMA phase lasted 14 days. Participants received three daily semi-random prompts (between 9:30–14:00, 14:00–18:30, and 18:30–23:00) to complete the momentary assessment through the data collection program Qualtrics. To prevent backfilling, participants were given sixty minutes to access the survey; after that period of time, the assessment was marked as missing. During the entire study, participants could contact a researcher on the team to resolve technical issues.

At the end of the study, participants were invited to return to the laboratory for a debriefing session. Participants who replied to at least 65% of the total EMA assessments received a monetary remuneration of 10 euros.

## 2.4 Statistical analyses

The datasets of the analyses and the R code are contained in an open-access file available on the OSF website at https://doi.org/10.17605/OSF.IO/TEUBR. The data analytic strategy followed three steps that are similar to the steps found in previous ecological momentary assessment studies on reciprocal influences between emotions and actions [29,36,53].

In an initial data preparation step, all the variables of interest were person-mean-centred to enable the examination of within-individual processes. Then, to analyse the relationships between variables assessed at two consecutive time points (t0 and t1), data were lagged. This meant deleting assessments that were not directly preceded or followed by another completed assessment ($n$ = 558). Consequently, each row of the data frame analysed contained participants' responses to two consecutive assessments (see the 'Data.csv' file at https://doi.org/10.17605/OSF.IO/TEUBR).

The first aim of the study was to examine the effect of the PE felt at a given time on the subsequent implementation of positive strategies. To this end, a series of linear mixed-effects models containing one random intercept per participant were estimated using maximum likelihood with the R lmerTest package [54]. Linear mixed-effects models were computed to take into account the hierarchical nature of the data. In this step, six models were computed (i.e., one per strategy). The dependent variable entered in each model was the change in the strategy of interest from t0 to t1, whereas the main independent variable was PE at t0. Two other independent variables were also included to neutralize their possible confounding effects: the use of each strategy at t0 and PE at t1. PE at t1 was included as a control variable. Not controlling for PE at t1 could produce a biased estimation of the effect of PE at t0 on the subsequent implementation of emotion regulation strategies. As PE at t1 was related to PE at t0 and strategy changes from t0 to t1, it could represent a confounding variable when attempting to determine the specific relationship between PE at t0 and strategy changes from t0 to t1. Therefore, to ensure that the effect of PE at t0 on strategy changes from t0 to t1 was not actually explained by PE at t1's relationships with both variables, we controlled for PE at t1

The second aim of the study was to examine the effect of positive emotion regulation strategies on subsequent PE level. To this end, one linear mixed-effects model was computed that contained PE at t1 as the dependent variable and change in the use of each strategy from t0 to t1 as independent variables. PE at t0 was also included as a control variable to neutralize the so-called regression towards the mean effect. Taken together, the analyses conducted to explore the first and second objectives of this study made it possible to analyse similar phenomena to those analysed in previous studies on reciprocal influences between emotions and actions in everyday life without resorting to their semi-retrospective assessment of strategy use (i.e., the effect of emotions at one time on the actions occurring between this time and a following time, and the effect of the actions performed within this time interval on concurrent emotional changes [e.g., 28,35,43]).

The third aim of the study was to explore whether the relationships between PE and positive emotion regulation significantly changed depending on strategy category. To this end, the dataset on which our analyses were based was restructured to obtain a data frame where each row contained a participant's responses to two consecutive assessments for one strategy category. In this restructured data frame, each pair of consecutive assessments completed consisted of three rows (one for the intensity of the use of attentional deployment strategies, one for the intensity of the use of cognitive change strategies and one for the intensity of the use of response modulation strategies; see the 'Data_ST.csv' file at https://doi.org/10.17605/OSF.IO/TEUBR). Then, the two types of linear mixed-effects models mentioned for the first and second aims of the study were computed again, but with slight modifications. A first model was designed to examine whether the effect of PE at t0 on the change in strategy use depended on the category of the strategy considered. The dependent variable was change in strategy use from t0 to t1, whereas the independent variables were PE at t0 and the interaction with the strategy category (i.e., a categorical variable with three modalities: attentional deployment, cognitive change, response modulation), with PE at t1 as a control variable. A second model was designed to examine the effects of change in strategy use on subsequent PE depending on

the category of strategy considered. This model included PE at t1 as the dependent variable, whereas the independent variables were change in strategy intensity, its interaction with the strategy category, and PE at t0.

## 3. Results

### 3.1 The influence of experienced positive emotions on positive emotion regulation

Descriptive statistics are shown in **Table 1**. They provide an initial general overview of the association between PE and positive emotion regulation.

The first aim of the study was to explore the effects of PE on positive emotion regulation. In a series of linear mixed-effects models (**Table 2**), we therefore examined how PE at t0 influenced changes in each of the strategies at t1, controlling for the use of each strategy at t0 and for PE at t1.

Results showed that the effects were all negative and significant. In other words, the less individuals felt positive emotions at t0, the more they tended to enhance the use of mindfulness ($b = -0.16$, $SE = 0.023$, $p < 0.001$), stimulus control ($b = -0.105$, $SE = 0.025$, $p < 0.001$), broadening ($b = -0.075$, $SE = 0.024$, $p < 0.01$), counting blessings ($b = -0.124$, $SE = 0.023$, $p < 0.001$), emotion expression ($b = -0.0647$, $SE = 0.025$, $p < 0.01$), and sharing ($b = -0.069$, $SE = 0.025$, $p < 0.01$) from this time to the next. Therefore, our results seem to confirm the hypothesis postulated by the hedonic flexibility principle, suggesting that the experience of low PE is likely to motivate individuals to subsequently increase the use of positive strategies in order to upregulate positive emotional states.

### 3.2 The influence of positive emotion regulation on experienced positive emotions

The second aim of the study was to explore the emotional outcomes of positive emotion regulation. We hypothesized that strategy change at t0 would predict PE at t1 and, more specifically, that an increase in the use of positive strategies would be associated with a greater experience of PE in the subsequent assessment.

**Table 2. Results of the six linear mixed-effects models predicting change in strategy use from PE at t0.**

| | Change in Mindfulness | | Change in Stimulus control | | Change in Broadening | | Change in Counting blessings | | Change in Emotion expression | | Change in Sharing | |
|---|---|---|---|---|---|---|---|---|---|---|---|---|
| | *b* | *SE* | *b* | *SE* | *b* | *SE* | *b* | *SE* | *b* | *SE* | *b* | *SE* |
| FIXED EFFECTS | | | | | | | | | | | | |
| PE (t0) | -.16*** | .023 | -.11*** | .025 | -.075** | .023 | -.124*** | .023 | -.065** | .025 | -.069** | .025 |
| Mindfulness (t0) | -.082** | .025 | .038 | .027 | .054* | .026 | .020 | .026 | .051 | .027 | .029 | .028 |
| Stimulus control (t0) | .012 | .023 | -.067** | .025 | .043 | .024 | .024 | .023 | .004 | .025 | .029 | .025 |
| Broadening (t0) | .069* | .028 | .024 | .03 | -.14*** | .028 | .096*** | .028 | .027 | .031 | .0097 | .031 |
| Counting blessings (t0) | .034 | .028 | .032 | .031 | .04 | .029 | -.124*** | .029 | -.019 | .03 | -.014 | .031 |
| Emotion expression (t0) | .038 | .023 | -.007 | .025 | .051* | .024 | .025 | .024 | -.072** | .025 | -018 | .025 |
| Sharing (t0) | .014 | .023 | .023 | .025 | -.016 | .024 | .027 | .024 | .054* | .025 | -.034 | .025 |
| PE (t1) | .55*** | .019 | .40*** | .02 | .48*** | .019 | .502*** | .019 | .044*** | .02 | .041*** | .025 |

*$p < .05$,

** $p < .01$,

***$p < .001$.

(PE: Positive emotions).

To test this hypothesis, a linear mixed-effects model was performed that included PE at t1 as the dependent variable (**Table 3**). Confirming our hypothesis, all the strategies were found to predict PE positively at t1, and, thus, an increase in the use of positive strategies at one time enhanced the experience of PE in the subsequent assessment (mindfulness: $b = 0.284$, $SE = 0.024$, $p < 0.001$; stimulus control: $b = 0.046$, $SE = 0.021$, $p < 0.05$; broadening: $b = 0.093$, $SE = 0.026$, $p < 0.001$; counting blessings: $b = 0.125$, $SE = 0.025$, $p < 0.001$; emotion expression: $b = 0.192$, $SE = 0.022$, $p < 0.001$; sharing: $b = 0.093$, $SE = 0.022$, $p < 0.001$), controlling for PE at t0 ($b = 0.21$, $SE = 0.016$, $p < .001$).

### 3.3 The moderating role of strategy category

Finally, we explored whether the association between PE and positive emotion regulation significantly changed depending on the strategy category. A first linear mixed-effects model investigated whether the strategy category affected the impact of PE at t0 on strategy use at t1 (**Table 4**). However, no significant interactions were observed.

We then examined whether the strategy category influenced the effect of change in strategy use on subsequent levels of PE (**Table 5**).

Interestingly, results revealed that the strategy category moderated the association between the change in strategy use and PE. More specifically, there was a significant interaction between change in strategy use and response modulation ($b = -0.161$, $SE = .027$, $p < .001$) and a close-to-significant trend in the interaction between change in strategy intensity and cognitive change ($b = -0.054$, $SE = .027$, $p = .059$). As **Fig 1** shows, the use of response modulation strategies to enhance PE was significantly less effective than the adoption of attentional deployment strategies.

## 4. Discussion

To date, although the use of strategies to regulate negative emotions has been extensively explored, the regulation of positive emotional states in everyday life has received little attention. The aim of the current study was to deepen our knowledge about PE and its underlying regulatory mechanisms. Overall, we showed that PE determines positive emotion regulation, which in turn affects subsequent levels of PE, thus confirming the existence of a reciprocal influence between momentary PE and positive emotion regulation.

**Table 3. Results of the linear mixed-effect model predicting PE at t1 from change in the use of each strategy at t0.**

|  | PE (t1) | | | |
|---|---|---|---|---|
|  | **b** | **SE** | **df** | **t** |
| FIXED EFFECTS |  |  |  |  |
| Change in mindfulness | 0.284*** | 0.024 | 2168 | 12.66 |
| Change in stimulus control | 0.046* | 0.021 | 2168 | 2.12 |
| Change in broadening | 0.093*** | 0.026 | 2168 | 3.73 |
| Change in counting blessings | 0.125*** | 0.025 | 2168 | 4.87 |
| Change in emotion expression | 0.192*** | 0.022 | 2168 | 5.47 |
| Change in sharing | 0.093*** | 0.022 | 2168 | 4.27 |
| PE (t0) | 0.21*** | 0.016 | 2168 | 12.85 |

*$p < .05$, ** $p < .01$,

***$p < .001$.

(PE = positive emotions).

**Table 4. Results of the linear mixed-effect model predicting the effect of positive emotions at t0 on change in strategy intensity at t1, moderated by strategy category.**

| | Change in use intensity (t1) | | | |
|---|---|---|---|---|
| | b | SE | df | t |
| FIXED EFFECTS | | | | |
| PE (t0) | -.174*** | .019 | 6546 | -9.31 |
| Cognitive change (vs. attentional deployment) | -.103*** | .024 | 6546 | -4.29 |
| Response modulation (vs. attentional deployment) | -.283*** | .025 | 6546 | -11.53 |
| Strategy intensity (t0) | -.010 | .011 | 6546 | -.91 |
| PE (t1) | .464*** | .011 | 6546 | 43.37 |
| PE (t0) * Cognitive change (vs. attentional deployment) | .028 | .025 | 6546 | 1.13 |
| PE (t0) * Response modulation (vs. attentional deployment) | .045 | .025 | 6546 | 1.83 |

*p < .05, ** p < .01,

***p < .001.

Attentional deployment represents the reference group. (PE: Positive emotions).

The first aim of the study was to explore the effects of PE on positive emotion regulation. The results showed that PE at t0 significantly predicted positive emotion regulation at t1 and, more specifically, that the less individuals felt PE at one time (t0), the more they tended to increase the use of positive strategies from this point to the next (from t0 to t1). Experiencing low levels of PE is, therefore, likely to shift one's efforts towards implementing strategies to reach a more positive emotional state. These findings are consistent with the hedonic flexibility principle [35,36], suggesting that individuals are likely to be motivated to upregulate PE as a consequence of low momentary affect. This regulatory mechanism might be seen as a highly adaptive process that can compensate for the lack of PE through an increased use of positive strategies, regardless of their nature (i.e., attentional, cognitive or behavioural). Moreover, the findings of the present study might also be understood in light of the affective baseline theory [55], which postulates the existence of a baseline functioning of an individual's affective system. According to this theory, although fluctuations around the home-base are the natural consequence of internal and external life events, affect is constantly brought back to the baseline by an attractive component consisting of regulatory mechanisms. Thus, the experience of low PE might encourage individuals to implement

**Table 5. Results of the linear mixed-effect model predicting the effect of change in strategy use intensity at t0 on PE at t1, moderated by strategy category.**

| | PE (t1) | | | |
|---|---|---|---|---|
| | b | SE | df | t |
| FIXED EFFECTS | | | | |
| Change in use intensity (t1) | .564*** | .021 | 6486.91 | 26.89 |
| Cognitive change (vs. attentional deployment) | .058* | .025 | 6439.02 | 2.36 |
| Response modulation (vs. attentional deployment) | .132*** | .024 | 6440.36 | 5.33 |
| PE (t0) | .252*** | .01 | 6538.48 | 4.06 |
| Change in use intensity (t1) * Cognitive change (vs. attentional deployment) | -.054 | .029 | 6524.83 | -1.89 |
| Change in use intensity (t1) * Response modulation (vs. attentional deployment) | -.161*** | .027 | 5933.7 | -5.92 |

*p < .05, ** p < .01,

***p < .001.

Attentional deployment represents the reference group. (PE: Positive emotions).

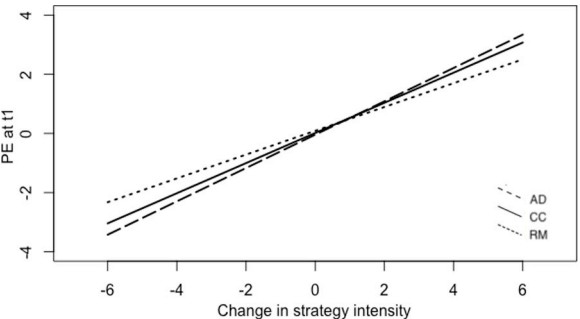

**Fig 1. Graphical representation of the effect of change in strategy use on subsequent positive emotions, moderated by strategy category (PE = positive emotions).**

strategies that induce a return to the baseline, which has been shown to be defined by a slightly positive valence [56].

In spite of being coherent with the hedonic theories, our results diverge from previous studies that showed increased implementation of positive strategies as a consequence of high levels of positive emotions [18,29,53]. A possible explanation for these divergent results might be found in the EMA design. All the previous studies adopted a momentary evaluation of the emotional state but a retrospective assessment of emotion regulation, asking participants to rate the strategies used since the last prompt (i.e., in the previous few hours). Thus, when analysing the effect of emotions at one point on subsequent strategy use, emotion regulation strategies were closer in time to the emotions than in our study, which assessed emotion regulation at exactly the same time as PE. In fact, there is evidence showing that soon after an emotion, mood-congruent processes take place, whereas mood-incongruent processes tend to follow [57]. Furthermore, the two theories suggested by the previous literature could be reconciled. The broaden-and-build theory states that the experience of PE encourages the building of adaptive resources and boosts the creation of coping skills, which in turn foster well-being. Accordingly, this 'resource-building process' might involve mechanisms that help to enhance affect when experiencing low PE and maintain a positive mood (i.e., consistent with hedonic theories), which in turn would promote psychological well-being and resilience in the long term.

The second aim of the study was to explore the unique emotional outcomes of positive emotion regulation. Confirming our hypothesis, an increase in the use of all six strategies resulted in enhanced PE in the subsequent assessment. In previous studies, trait savoring was found to be associated with greater happiness and well-being [20], whereas state savoring was shown to predict increased positive emotions [21]. Therefore, these results are consistent with the previous literature and support the adaptive role of positive emotion regulation in emotional well-being.

Finally, our third aim was to investigate whether the reciprocal influence between PE and positive emotion regulation changed significantly depending on the intrinsic nature of the strategies. The results showed that the strategy category significantly moderated the association between the change in strategy use and subsequent PE. More specifically, the use of response modulation strategies (e.g., sharing and emotional expression) was significantly less effective than the adoption of attentional deployment strategies and produced a less important increase in subsequent PE. Sharing positive experiences has been shown to improve one's perception in the eyes of others, leading to increased self-esteem [58] and life satisfaction [20]. This strategy might, therefore, indirectly increase PE by mainly targeting other dimensions of an

individual's well-being. Furthermore, the benefits of sharing have been shown to depend on how the recipient responds to the news (actively/constructively or passively/destructively) [58], which could further justify the mitigated effects of this strategy on momentary PE found in our study. In contrast, emotional expression refers to the verbal or nonverbal expression of an ongoing emotion [59], which makes it possible to rapidly and adaptively react to environmental threats and opportunities [60]. Emotional expression may foster PE, especially in the short term (i.e., soon after the emotion is produced), thus showing reduced effects in the long term. As suggested in a previous study [20], positive emotion regulation might not only increase only PE. Instead, each strategy may target different dimensions of the person's emotional well-being, thus involving different emotional outcomes. However, further studies are needed to disentangle the unique emotional consequences of positive emotion regulation.

Although this study sheds new light on the mechanisms underlying the experience of PE, we acknowledge several limitations that could be addressed by future research.

First, our study involved a sample of 85 healthy undergraduate individuals. Future studies are needed to explore the reciprocal influence between PE and positive emotion regulation in a more diverse sample.

Second, we excluded participants who presented moderate-to-severe depressive symptoms. It is possible that the patterns observed in the present study cannot be extended to samples of patients suffering from an emotional disorder [61], who are typically prone to dampening rather than savoring PE [48,49]. Reasonably, an abnormal functioning of this mechanism might be observed in this population, which could be defined by a lack of motivation or capacity to implement positive strategies despite experiencing low PE, or by reduced efficacy in using positive strategies to increase PE levels. Future studies should confirm this hypothesis.

Third, our study specifically focused on PE, without studying the role of negative affect on positive emotion regulation. A growing body of evidence shows that positive and negative affect do not lie on opposite ends of a bipolar scale; instead, they can be experienced simultaneously [62,63]. In our study, we found that the experience of low PE was associated with a greater use of positive strategies, which might suggest that upregulating PE also serves as a mechanism to repair mood [64]. Nevertheless, the absence of a variable assessing momentary negative emotions keeps us from confirming this hypothesis, which should be addressed in future studies.

Fourth, the daily EMA included the assessment of only six positive strategies. On the one hand, there is evidence that people's repertoire for dealing with PE includes a wider range of strategies that were not explored in this study [65]. On the other hand, the use of maladaptive strategies in response to positive states (e.g., dampening) was not taken into consideration, thus limiting the findings of the present study to the mechanisms underlying the upregulation of PE.

Finally, the use of ad hoc single items to assess a multifaceted construct such as emotion regulation might not fully capture the complexity of this process. In addition to the fact that the use of ad hoc items is common in EMA studies [see for example, 25,30,32,38], the validated questionnaires available to assess positive emotion regulation mainly measure an individual's tendency to savour positive emotions (see for example [11,66,67]), rather than measuring to what extent specific strategies are adopted in response to a specific stimulus (e.g., state emotion regulation). The lack of validated measures for the assessment of momentary positive regulation led us to create our own single items. Moreover, there is evidence that long EMA questionnaires usually lead to higher perceived burden [68], which further supports the decision to include only a few items to assess emotion regulation. Indeed, the inclusion of a broader set of items could have resulted in decreased compliance and increased participant burden, thus affecting the quality of the data collected. As Trull and Ebner-Premier [69] recently stated,

EMA is still a field with several methodological aspects that remain unclear. Accordingly, there is the need to expand and improve this research field further by, for instance, creating validated measures to be used in EMA designs or developing rigorous guidelines that guides researchers in the design of EMA studies.

Despite these limitations, our research adds to the previous literature by extending our knowledge about PE and the underlying regulatory mechanisms. More specifically, we showed that low levels of PE determine an increase in the use of strategies to upregulate PE, which in turn results in a better mood.

Although further studies are needed to confirm these findings, our study sheds new light on the importance of PE for emotional well-being, and it opens up new avenues to understand the dysfunctional regulation of positive emotional states in emotional disorders.

## Author Contributions

**Conceptualization:** Desirée Colombo.

**Formal analysis:** Jean-Baptiste Pavani.

**Investigation:** Desirée Colombo.

**Methodology:** Jean-Baptiste Pavani.

**Supervision:** Jean-Baptiste Pavani, Azucena Garcia-Palacios, Cristina Botella.

**Writing – original draft:** Desirée Colombo.

**Writing – review & editing:** Javier Fernandez-Alvarez, Cristina Botella.

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
