## [Decision Letter · Decision Letter 0]

26 Oct 2020

PONE-D-20-28135

Savoring the present: The reciprocal influence between positive emotions and positive emotion regulation in everyday life

PLOS ONE

Dear Dr. Colombo,

Thank you for submitting your manuscript to PLOS ONE. After careful consideration, we feel that it has merit but does not fully meet PLOS ONE’s publication criteria as it currently stands. Therefore, we invite you to submit a revised version of the manuscript that addresses the points raised during the review process.

I have now received 2 reviews of your manuscript from experts in the field. Both reviewers identified several strengths in your paper, including the potential contribution of the research to the field. However, the reviewers also note several areas of weakness where more information and greater elaboration of the theoretical framework underlying the research would be beneficial. In particular, both reviewers note issues with how Gross’s process model of emotion regulation has been applied in the current research and could be better articulated. Reviewer 1 in particular highlights issues with the way in which the emotion regulation strategies have been described including a lack of a clear fit with Gross’s model.  

Although I will not repeat all of the reviewer’s recommendations for improvement here, I do believe that their concerns (and especially those of Reviewer 2) are substantial enough that the paper cannot be accepted in its present form.

However, it may be possible that with careful attention to their comments, a substantially revised paper could be considered for publication. Given this, I would like to invite you to make substantial revisions in line with the reviewers’ comments and resubmit a revised version of the manuscript. Please include a cover letter detailing how you have dealt with each of the comments.

Thank you for considering PLOS ONE as an outlet for your work and I look forward to receiving your revised manuscript.

We look forward to receiving your revised manuscript.

Kind regards,

Fuschia M. Sirois, PhD

Academic Editor

PLOS ONE

Journal Requirements:

Reviewers' comments:

Reviewer's Responses to Questions

**Comments to the Author**

1. Is the manuscript technically sound, and do the data support the conclusions?

Reviewer #1: Partly

Reviewer #2: Yes

2. Has the statistical analysis been performed appropriately and rigorously? 

Reviewer #1: Yes

Reviewer #2: Yes

3. Have the authors made all data underlying the findings in their manuscript fully available?

Reviewer #1: Yes

Reviewer #2: Yes

4. Is the manuscript presented in an intelligible fashion and written in standard English?

Reviewer #1: Yes

Reviewer #2: No

5. Review Comments to the Author

Reviewer #1: My biggest concern with the current study is that the factorial and construct validity of the use of ad-hoc single items to assess the implementation of positive strategies. Although in the discussion, the authors state that “the use of ad-hoc single items to assess a multifaceted construct like emotion regulation might have not fully captured the complexity of this process”, the explanation that “no validated questionnaire assessing state positive regulation has been so far developed” is not sufficient.

Second, the classification of positive strategies may appear arbitrary. For example, the authors categorised two strategies, broadening and count blessing, into the “cognitive change” group. The item that used to assess “broadening” strategy is that “I’m thinking about all the good things that I have and that are happening in my life as well. ” and the item that used to assess count blessing is that “I’m thinking about how lucky I am to live this moment and feel so good. However, within the process model of emotion regulation (Gross, 1998, 2015), the core element of cognitive change is modifying the meaning of the situation to alter an emotional response. Theoretically, they are different strategies.

Third, as a measure of the momentary intensity of positive emotions, participants were asked to rate 7 adjectives on a 1-5 Likert, including “happiness, amusement, hope, serenity, excitement, pride, gratitude”. However, the reasons for choosing these adjectives are not clarified.

Fourth, the authors mention in their discussion that “upregulating PE may therefore serve as a mechanism to repair mood and reduce the experience of unpleasant affective states”. This conclusion is arbitrary due to it may be based on the assumption that PA and NA are on two opposite ends of a bipolar scale (i.e., People can be experiencing one type of affect to a certain degree but not the other at the same time). Indeed, there is a growing body of evidence that positive and negative affect can co-occur simultaneously (e.g., Berrios, Totterdell, & Kellett, 2015; Larsen & McGraw, 2011).

Fifth, the authors mention in the introduction “people who frequently adopt savouring strategies, for instance, show higher happiness, greater positive affect and enhanced emotional well-beings ” (Page 4, Paraphrase 2). However, the meaning of “savouring strategies” did not clearly defined at any other point in the manuscript.

Reviewer #2: This manuscript summarizes an interesting study on an understudied topic. Although there are several notable limitations and suggestions for the manuscript (listed below), the study, particularly objective 1, may offer a contribution to the literature.

1. The introduction mentions multiple ways people can regulate emotions (e.g., Gross’s process model and Bryant’s conceptualization of savoring past, present, future), but these aren’t well integrated or applied to the current study. It is recommended that the authors are more explicit about how frameworks relate and inform the current study.

2. More justification should be provided about how and why the use of specific positive emotion regulation strategies should predict more positive emotions over time. Authors should explain what strategies they examine and why these may be useful to increase positive affect. They should mention that they only include ‘adaptive’ strategies given that maladaptive ones (e.g., dampening, suppressing, etc.) may curtail positive affect. Authors should incorporate other longitudinal studies that have shown that savoring types of ER strategies increase positive emotions over time. Examples include: Gentzler, Morey, Palmer, & Yi, 2012; Hurley & Kwon, 2013; Jose, Lim, Bryant, 2012; Langston, 1994.

3. Researchers should justify their sample size of 85 undergraduates and their chosen time frame for the study (3 measurements a day for two weeks). There is no rationale for either. It also isn’t clear if they eliminated anyone due to not completing a minimum number of ESM assessments.

4. Were the emotion scales labeled (the 1 and 5 endpoints or each number)?

5. The ER questions need additional details so the reader can understand what was done and why. Examples should be provided. What is “stimulus control” as a ER strategy? Why are difference scores computed and what do Time 0 to Time 1 represent? (A little more information is given in the results, but it should be defined in the method given it is first mentioned there.) There are existing measures of positive emotion regulation in the literature so better explaining the benefits of these items may be helpful.

6. Descriptives for the emotion and ER rates should be provided (maybe in Table 1).

7. In Table 2, the notations to flag significant associations are included but they don’t actually use them in the table. Instead they write out the p-values. However, given the table is very hard to read, using asterisks is recommended.

8. Researchers tested additional hypotheses not outlined in the introduction. These should not take the reader by surprise, so these should be mentioned and justified earlier.

9. The benefits of sharing positive experiences also depend on how the recipients respond to the news (Shelly Gable’s research). Therefore, recipients’ supportive v. less supportive responses may mitigate its positive effects sometimes.

10. Additional limitations include not including maladaptive responses to PA (e.g., dampening) and negative affect. When people are lower on PA, do they also do more ineffective ER? It seems likely that they do not solely use the types of ER strategies that happened to be assessed here.

11. There are many grammatical issues with this paper, so these would need to be corrected.

6. PLOS authors have the option to publish the peer review history of their article (what does this mean?). If published, this will include your full peer review and any attached files.

Reviewer #1: No

Reviewer #2: No

---

## [Author Response · Author response to Decision Letter 0]

17 Nov 2020

Reviewer #1

My biggest concern with the current study is that the factorial and construct validity of the use of ad-hoc single items to assess the implementation of positive strategies. Although in the discussion, the authors state that “the use of ad-hoc single items to assess a multifaceted construct like emotion regulation might have not fully captured the complexity of this process”, the explanation that “no validated questionnaire assessing state positive regulation has been so far developed” is not sufficient.

Thank you for raising this point. As mentioned in the discussion, the use of single items to assess positive strategies might represent a criticism of the study. However, to the best of our knowledge, there aren’t validated items assessing state (e.g., momentary) positive regulation. The available questionnaires usually assess positive emotion regulation in a more general way (see for example the Response Style to Positive Affect Scale, which assesses emotion-focused or self-focused savoring; the Savoring Belief Inventory, which only explores savoring in terms of anticipating, experiencing or recalling positive states; or the Emotion Regulation Profile-Revised, which consists of narratives rather than items). Additionally, most of the available emotion regulation questionnaires relies on the idea that emotion regulation is a trait of a person, thus assessing emotion regulation as a cross-situational tendency. Only a few questionnaires have been developed to assess state emotion regulation (i.e., the momentary adoption of strategies in relation to a specific event, which would fit better with the use of EMA), but only in relation to negative emotions (see for example the Brief State Rumination Inventory, Marchetti et al., 2018; or the State Difficulties in Emotion Regulation Scale, Lavender et al., 2017). Finally, there is evidence showing that longer EMA questionnaires usually imply higher perceived burden and lower compliance to the experimental protocol, which further support our choice to use single items. Thus, integrating a full questionnaire of positive regulation in the EMA would have been counterproductive.

Broadly talking, the use of single items isn’t just a limitation of our study, but a very important issue of the EMA literature. Indeed, almost all EMA studies rely on the use of ad-hoc items, thus raising important issues regarding their construct validity. We acknowledge this criticism, and we recognise that there is the need to further expand this research field by creating validated measures to be used in momentary assessments. These efforts have been already made in other research fields. An example is the Distress Thermometer, a single measure which has been developed to assess “unpleasant experience of a mental, physical, social, or spiritual nature”. Similarly, validated items to assess emotion regulation should be created in the near future.

As the reviewer will see, we have tried to incorporate the aforementioned considerations in the manuscript and to better justify the use of single items (from line 484). 

Second, the classification of positive strategies may appear arbitrary. For example, the authors categorised two strategies, broadening and count blessing, into the “cognitive change” group. The item that used to assess “broadening” strategy is that “I’m thinking about all the good things that I have and that are happening in my life as well. ” and the item that used to assess count blessing is that “I’m thinking about how lucky I am to live this moment and feel so good. However, within the process model of emotion regulation (Gross, 1998, 2015), the core element of cognitive change is modifying the meaning of the situation to alter an emotional response. Theoretically, they are different strategies.

This is a very interesting point, thank you for the comment. Our study was based on Gross’ model of emotion regulation (2015), which was originally thought for the regulation of negative affective states. Recently, Quoidbach et al. (Quoidbach, Mikolajczak and Gross, 2015) applied Gross’ model to explain and understand the regulation of positive emotions. Accordingly, our study was inspired by Quoidbach et al.’s theory.

In their model, Quoidbach et al. suggest counting blessings (“Are you realizing how lucky you are or are you taking the situation for granted?”) as an example of cognitive change in response to positive emotions, which is described as the attempt to “change how one appraises the situation in order to alter its emotional significance” by “appraising the situation as a special moment” or “by increasing how valuable positive events appear”. We therefore decided to include count blessing as a cognitive change strategy. Furthermore, we included broadening as a further example of cognitive change because it does reflect a way of appraising a positive situation and generalizing it to one’s positive things in life, which is consistent with the aforementioned definition. 

Regarding the second category, Quoidbach et al. define attentional deployment as the attempt to influence positive emotions by specifically directing the attention within a situation (“What do you purposefully pay attention to during your date night?”). Accordingly, we decided to include mindfulness (I’m trying to be focused on the present and to concentrate on how good I feel) and stimulus control (I’m trying to avoid all negative thoughts and stressors in order to focus and take the most of my positive emotions) as two strategies that reflects the attempt to modify one’s attentional focus in order to take the most out of a positive emotion. 

Finally, Quoidbach et al. describe response modulation as the attempt to “influence physiological, experiential, or behavioral responding as directly as possible” and suggest social sharing and emotional expression as two examples of this category (“Are you laughing and sharing your positive feelings with your partner or, conversely, hiding your emotions?”). We therefore decided to include both strategies to represent this category.

We do completely agree with the reviewer that clearer definitions of the categories were missing. We have now substantially modified the manuscript in order to clarify these points Moreover, and consistent with reviewer#2’s suggestion to explain why we expected those strategies to increase positive emotions, we have also tried to further justify the selection of the six specific strategies included in the study (from line 192; from line 198).

Third, as a measure of the momentary intensity of positive emotions, participants were asked to rate 7 adjectives on a 1-5 Likert, including “happiness, amusement, hope, serenity, excitement, pride, gratitude”. However, the reasons for choosing these adjectives are not clarified.

Thank you for raising this issue. We have now included more details regarding the selection of the seven positive emotions. More specifically, a previous study found that the regulation of positive emotions is likely to affect both active (e.g. happiness) and deactive (e.g., calmness) positive affect (Nezleck et al., 2008). Accordingly, we selected the seven emotions in order to include both low- (hope, serenity, gratitude) and high-arousal (happiness, amusement, excitement, pride) positive emotions. This information has been now included in the manuscript (from line 184).

Fourth, the authors mention in their discussion that “upregulating PE may therefore serve as a mechanism to repair mood and reduce the experience of unpleasant affective states”. This conclusion is arbitrary due to it may be based on the assumption that PA and NA are on two opposite ends of a bipolar scale (i.e., People can be experiencing one type of affect to a certain degree but not the other at the same time). Indeed, there is a growing body of evidence that positive and negative affect can co-occur simultaneously (e.g., Berrios, Totterdell, & Kellett, 2015; Larsen & McGraw, 2011).

We completely understand the reviewer’s point. As correctly mentioned by the reviewer, and consistent with an extensive literature, experiencing low levels of positive emotions does not imply experiencing high levels of negative affect. Indeed, the fact that we only assessed momentary PE could be considered a limitation of the study, as it does not allow to fully comprehend the reciprocal influence between positive emotion regulation and momentary affect (both positive and negative). We therefore decided to move this statement to the limitations of the study and to discuss it in more details. More specifically, we state that, despite the results might support the mood-repair theory, such conclusion can’t be confirmed because of the lack of information regarding momentary negative affect (from line 470).

Fifth, the authors mention in the introduction “people who frequently adopt savouring strategies, for instance, show higher happiness, greater positive affect and enhanced emotional well-beings ” (Page 4, Paraphrase 2). However, the meaning of “savouring strategies” did not clearly defined at any other point in the manuscript.

Thank you very much for pointing out this issue. We have now included a short definition of “savoring strategies” in the introduction (line 96).

Reviewer #2

1. The introduction mentions multiple ways people can regulate emotions (e.g., Gross’s process model and Bryant’s conceptualization of savoring past, present, future), but these aren’t well integrated or applied to the current study. It is recommended that the authors are more explicit about how frameworks relate and inform the current study.

Thank you for the suggestion. The introduction has been significantly revised in order to conceptualize in a clearer way the different frameworks related to the regulation of positive emotion regulation. More specifically, we have now underlined that, despite emotion regulation can occur before (anticipating), during (experiencing) or after (recalling) the generation of a positive emotion, the focus of our study is on the application of Gross’ extended model to the regulation of ongoing positive emotions suggested by Quoidbach et al. (thus, experiencing) (from line 89, from line 145).

2. More justification should be provided about how and why the use of specific positive emotion regulation strategies should predict more positive emotions over time. Authors should explain what strategies they examine and why these may be useful to increase positive affect. They should mention that they only include ‘adaptive’ strategies given that maladaptive ones (e.g., dampening, suppressing, etc.) may curtail positive affect. Authors should incorporate other longitudinal studies that have shown that savoring types of ER strategies increase positive emotions over time. Examples include: Gentzler, Morey, Palmer, & Yi, 2012; Hurley & Kwon, 2013; Jose, Lim, Bryant, 2012; Langston, 1994.

We completely agree with the reviewer’s point. The rationale behind the selection of the strategies was not adequately developed. In order to address this issue, two main changes were made. 

First, and according to reviewer#1 suggestion, a more detailed explanation about the classification of positive strategies has been provided, which was based on the work by Quoidbach et al. (Quoidbach, Mikolajczak and Gross, 2015), who applied Gross’ emotion regulation model to the regulation of positive emotions (from line 192; from line 198).

- Cognitive change: In this work, the authors provide count blessing (“Are you realizing how lucky you are or are you taking the situation for granted?”) as an example of cognitive change in response to positive emotions, which is described as the attempt to “change how one appraises the situation in order to alter its emotional significance” by “appraising the situation as a special moment” and suggesting that “positive emotions do not depend on situations per se as much as they depend on the way individuals interpret these situations”. We therefore decided to include count blessing as a cognitive change strategy. Furthermore, we included broadening as a further example of cognitive change because it does reflect a way of appraising a positive situation as a special moment, which is consistent with the aforementioned definition. Moreover, broadening has been shown to be one of the most used strategies in response to positive emotions (Heiy and Cheavens, 2014).

- Attentional deployment: The authors define attentional deployment as the attempt to influence positive emotions by specifically directing our attention within a situation (“What do you purposefully pay attention to during your date night?”). Accordingly, we decided to include mindfulness (I’m trying to be focused on the present and to concentrate on how good I feel) and stimulus control (I’m trying to avoid all negative thoughts and stressors in order to focus and take the most of my positive emotions) as two strategies that reflects the attempt to modify one’s attentional focus in order to take the most out of a positive emotion.

- Response modulation: The authors define response modulation as the attempt to “influence physiological, experiential, or behavioral responding as directly as possible” and suggest social sharing and emotional expression as two examples of this category (“Are you laughing and sharing your positive feelings with your partner or, conversely, hiding your emotions?”). We therefore decided to include both strategies to represent this category.

Second, we have now included more information to justify the selection of the six strategies and to support their potential role on momentary affect (from line 198).

Besides, we have mentioned the lack of maladaptive strategies within the EMA assessment as a further limitation of the study (see response to comment 10) (from line 477). Finally, we have included the main findings of the studies suggested by the reviewer to further support the important emotional outcomes of positive emotion regulation (from line 95).

3. Researchers should justify their sample size of 85 undergraduates and their chosen time frame for the study (3 measurements a day for two weeks). There is no rationale for either. It also isn’t clear if they eliminated anyone due to not completing a minimum number of ESM assessments.

Thank you for raising these criticisms. Regarding the first issue, we decided the sample size by taking into account the sample sizes used in previous EMA studies on emotion regulation (see for example Pavani et al., 2015; Brans et al., 2013) (line 169). Regarding study design, we reviewed the past EMA literature to decide the best sampling frequency to adopt. However, no general rule was found, considering the incredibly huge variability among studies assessing ER through EMA protocols, ranging from once per day (see for example Nezlek et al. 2008) to ten times per day (see for example Kuppens et al. 2010). Because of the lack of shared guidelines, we decided to rely on the study by Heiy and colleagues (2015), which was used as a reference for the design of our study (especially for the ER items) and which for sure represents an important work within the field. Moreover, our previous studies adopting the same sampling frequency supports the feasibility of such design, which is reflected by the low burden and high compliance of participants (80.47%, Colombo et al., 2020a; 77.8%, Colombo et al., 2020b). Accordingly, more details about the selection of the sampling frequency have been included in the manuscript (from line 179).

Finally, we did not put a threshold on compliance levels in order to include or exclude participants. This information has been now included in the manuscript (line 182).

4. Were the emotion scales labeled (the 1 and 5 endpoints or each number)?

The endpoints of the Likert scales used to assess positive emotions were labelled (1=not at all; 5=a lot). We have now included this information in the manuscript (line 184).

5. The ER questions need additional details so the reader can understand what was done and why. Examples should be provided. What is “stimulus control” as a ER strategy? 

We do completely agree that the selection of the strategies was not adequately detailed. As the reviewer will see, we have now clarified this issue. More specifically, and following reviewer#1 suggestion, we have now included an explanation of each ER category (attentional deployment, cognitive change, response modulation), which helped us introduce each of the positive strategy of our study. Furthermore, further references have been included in order to support their selection (from line 198).

Why are difference scores computed and what do Time 0 to Time 1 represent? (A little more information is given in the results, but it should be defined in the method given it is first mentioned there.) 

In the revised manuscript, we have now included a clearer explanation about the change scores we calculated for each emotion regulation strategy (from line 229).

There are existing measures of positive emotion regulation in the literature so better explaining the benefits of these items may be helpful 

Thank you for raising this point. As mentioned in the discussion, the use of single items to assess positive strategies might represent a criticism of the study. However, to the best of our knowledge, there aren’t validated items assessing state (e.g., momentary) positive regulation. The available questionnaires usually assess positive emotion regulation in a more general way (see for example the Response Style to Positive Affect Scale, which assesses emotion-focused or self-focused savoring; the Savoring Belief Inventory, which only explores savoring in terms of anticipating, experiencing or recalling positive states; or the Emotion Regulation Profile-Revised, which consists of narratives rather than items). Additionally, most of the available emotion regulation questionnaires relies on the idea that emotion regulation is a trait of a person, thus assessing emotion regulation as a cross-situational tendency. Only a few questionnaires have been developed to assess state emotion regulation (i.e., the momentary adoption of strategies in relation to a specific event, which would fit better with the use of EMA), but only in relation to negative emotions (see for example the Brief State Rumination Inventory, Marchetti et al., 2018; or the State Difficulties in Emotion Regulation Scale, Lavender et al., 2017). Finally, there is evidence showing that longer EMA questionnaires usually imply higher perceived burden and lower compliance to the experimental protocol, which further support our choice to use single items. Thus, integrating a full questionnaire of positive regulation in the EMA would have been counterproductive.

Broadly talking, the use of single items isn’t just a limitation of our study, but a very important issue of the EMA literature. Indeed, almost all EMA studies rely on the use of ad-hoc items, thus raising important issues regarding their construct validity. We acknowledge this criticism, and we recognise that there is the need to further expand this research field by creating validated measures to be used in momentary assessments. These efforts have been already made in other research fields. An example is the Distress Thermometer, a single measure which has been developed to assess “unpleasant experience of a mental, physical, social, or spiritual nature”. Similarly, validated items to assess emotion regulation should be created in the near future.

As the reviewer will see, we have tried to incorporate the aforementioned considerations in the manuscript and to better justify the use of single items (from line 484). 

6. Descriptives for the emotion and ER rates should be provided (maybe in Table 1).

The descriptives for the emotions and ER rates have been included in Table 1, as suggested by the reviewer.

7. In Table 2, the notations to flag significant associations are included but they don’t actually use them in the table. Instead they write out the p-values. However, given the table is very hard to read, using asterisks is recommended.

Thank you for pointing out this mistake. We have now corrected the table by including the asterisks and deleting the p-values. We also modified the other tables, consistently.

8. Researchers tested additional hypotheses not outlined in the introduction. These should not take the reader by surprise, so these should be mentioned and justified earlier.

We completely agree with the reviewer. Indeed, the moderation of strategy category on the reciprocal influence between positive ER and PE was not mentioned among the hypotheses. We have now included it (from line 16\\).

9. The benefits of sharing positive experiences also depend on how the recipients respond to the news (Shelly Gable’s research). Therefore, recipients’ supportive v. less supportive responses may mitigate its positive effects sometimes.

This is a very interesting point, which further explains the results observed in relation to the use of social sharing. We have now included it in the discussion of the results (from line 448).

10. Additional limitations include not including maladaptive responses to PA (e.g., dampening) and negative affect. When people are lower on PA, do they also do more ineffective ER? It seems likely that they do not solely use the types of ER strategies that happened to be assessed here.

Thank you for the suggestion. We have now underlined the lack of items to assess maladaptive strategies and negative affect as two further limitations of the present study. Regarding the latter, one of the reviewers suggested that, the lack of measures about negative affect, does not allow to conclude that “upregulating PE may serve as a mechanism to repair mood and reduce the experience of unpleasant affective states”. Indeed, the fact that we only assessed momentary PE levels does not allow to fully comprehend the reciprocal influence between positive emotion regulation and momentary affect (both positive and negative). We therefore decided to move this statement to the limitations of the study. More specifically, we state that, despite the results might support the mood-repair theory, such conclusion can’t be confirmed because of the lack of information regarding momentary negative affect (from line 470).

11. There are many grammatical issues with this paper, so these would need to be corrected.

The manuscript has been edited by an English-speaking native in order to correct grammatical issues.

---

## [Decision Letter · Decision Letter 1]

10 Feb 2021

PONE-D-20-28135R1

Savoring the present: The reciprocal influence between positive emotions and positive emotion regulation in everyday life

PLOS ONE

Dear Dr. Colombo,

Thank you for submitting your manuscript to PLOS ONE. After careful consideration, we feel that it has merit but does not fully meet PLOS ONE’s publication criteria as it currently stands. Therefore, we invite you to submit a revised version of the manuscript that addresses the points raised during the review process.

Reviewer 2 notes that although many of the issues raised in the last review were addressed, some still remain. As well this reviewer notes that additional issues have been introduced in making some of the revisions. The Reviewer has provided a very thoughtful and thorough list of these issues and detailed the areas needed for further improvement. Please read through these carefully and respond to each, and how or if you will address them in your revision cover letter.

We look forward to receiving your revised manuscript.

Kind regards,

Fuschia M. Sirois, PhD

Academic Editor

PLOS ONE

Reviewers' comments:

Reviewer's Responses to Questions

**Comments to the Author**

1. If the authors have adequately addressed your comments raised in a previous round of review and you feel that this manuscript is now acceptable for publication, you may indicate that here to bypass the “Comments to the Author” section, enter your conflict of interest statement in the “Confidential to Editor” section, and submit your "Accept" recommendation.

Reviewer #1: All comments have been addressed

Reviewer #2: (No Response)

2. Is the manuscript technically sound, and do the data support the conclusions?

Reviewer #1: Yes

Reviewer #2: Partly

3. Has the statistical analysis been performed appropriately and rigorously? 

Reviewer #1: Yes

Reviewer #2: I Don't Know

4. Have the authors made all data underlying the findings in their manuscript fully available?

Reviewer #1: Yes

Reviewer #2: Yes

5. Is the manuscript presented in an intelligible fashion and written in standard English?

Reviewer #1: Yes

Reviewer #2: No

6. Review Comments to the Author

Reviewer #1: (No Response)

Reviewer #2: The authors were somewhat responsive to earlier concerns. However, certain points were not addressed as expected or the changes resulted in new problems. Although I believe this paper has several strengths and could contribute to the literature, the remaining issues dampen my enthusiasm but could be addressed in the future.

Introduction

1. Terms are not defined in the introduction when first mentioned – capitalizing, savoring, AD, CC, RM. It doesn’t make sense to wait until the method to cite literature explaining what the strategies are and what they relate to.

2. Relatedly they do not justify the particular emotion regulation strategies that they focus on. Some of this text (justifying why these strategies would relate to more PE) is included in the discussion, but it would be helpful to have this information in the introduction and state they’re excluding maladaptive ER.

3. This sentence needs more explanation given this study doesn’t study mindfulness, self-esteem, and autonomy: “ Furthermore, intense use of mindfulness has been shown to predict higher levels of daily autonomy (24), whereas positive reappraisal has been associated with greater self-esteem and well-being (25).” Although there is a mindfulness type of item as ER, readers don’t know that until the method, so the sentence appears mostly unrelated.

Method and results

4. A major problem in the paper remains unclear description and treatment of ER strategies.

a. If the individual strategies are being collapsed into categories, why isn’t this done consistently? Some analyses examine ER as individual strategies (all 6) and others include combined strategies (AD, CC, RM) but there is no rationale for the switch back and forth.

b. The use of acronyms for AD, CC, and RM are confusing as they are not well known. Can these be written out?

c. It is not clear from Table 1 that authors are correlating scores that define other scores (e.g., expressing and sharing are RM, so that’s why the correlations are .90). This should be explained in a note or authors should pick one (either the aggregate or individual scores) and use one throughout the paper.

5. Do authors test for significant differences between strategy use? It is not clear based on how it is written and it’s not clear how much value this text adds.

6. Why is T1 PE controlled for when predicting ER strategies from PE T0? Authors should have a good rationale for including it and explain it in the paper (or instead not include it as a covariate).

7. Conversely, it seems that T0 PE should be included as a covariate when predicting PE T1 from ER. Was that done? Where all 8 strategies run in separate models? Why isn’t AD included in Tables 3 and 4 and corresponding analyses? More information should be included about the models testing these hypotheses, and potentially including analyses in a table given they are major findings of the paper.

8. The rationale for the analysis in Table 4 is not provided and the analyses and results are not clear. It is recommended these are removed or set up as a hypotheses and clarified. The authors said they added the analyses in Table 3 and 4 as hypotheses, but the one sentence added at the end of the introduction is not clear in terms of corresponding to Table 4.

Discussion

9. The description of some of the results is not summarized as clearly as it could. For example: “Interestingly, the tendency to adopt positive strategies to increase PE was less effective when using RM strategies (e.g., sharing and emotional expression).”

a. Was this difference significant? What was it less effective than?

10. In general, it is hard to see breaking of paragraphs, but it appears the last paragraph of the paper goes on for two pages. They should break it into two paragraphs (last 3 sentences could be on their own).

11. Several typos are present. For example:

a. Langstone instead of Langston

b. In the measures paragraphs there are multiple typos, and use of so many parentheses is very hard to follow.

7. PLOS authors have the option to publish the peer review history of their article (what does this mean?). If published, this will include your full peer review and any attached files.

Reviewer #1: No

Reviewer #2: No

---

## [Author Response · Author response to Decision Letter 1]

30 Mar 2021

Reviewer #1

The authors were somewhat responsive to earlier concerns. However, certain points were not addressed as expected or the changes resulted in new problems. Although I believe this paper has several strengths and could contribute to the literature, the remaining issues dampen my enthusiasm but could be addressed in the future.

We are very pleased to know that the reviewer believes that the manuscript might represent an important contribution. Also, we are very grateful for the insightful points outlined and we have enthusiastically tried to address all concerns. In our resubmitted manuscript, we have highlighted areas of substantive changes in yellow, so that the editor and the reviewer can easily identify them. Material that has been moved around from one place to another is in bolded green font.

Terms are not defined in the introduction when first mentioned – capitalizing, savoring, AD, CC, RM. It doesn’t make sense to wait until the method to cite literature explaining what the strategies are and what they relate to.

Thank you very much for the suggestion. We have now included in the manuscript the definitions of the terms mentioned by the reviewer (see for example line 66 or line 100). Furthermore, we have moved the literature supporting the six positive strategies from the methods to the introduction (from line 164). We do agree that it is more consistent to provide this information to the reader when introducing our study.

Relatedly they do not justify the particular emotion regulation strategies that they focus on. Some of this text (justifying why these strategies would relate to more PE) is included in the discussion, but it would be helpful to have this information in the introduction and state they’re excluding maladaptive ER.

We have now moved to the introduction the paragraphs justifying the selection of the six strategies and discussing their association with PE (“The current study”) (from line 164), and we have included a more detailed explanation about the rational used to select categories and strategies. Categories were selected based on the review by Quoidbach et al. (2015), who points out that “Regarding short-term increases in positive emotions, our review found that attentional deployment, cognitive change, and response modulation strategies have received the most empirical support, whereas more work is needed to establish the effectiveness of situation selection and situation modification strategies” (from line 153). Strategies were selected in order to explore these 3 categories of emotion regulation (two strategies for each category) and according to the literature supporting the association between these specific strategies and PE. Furthermore, and as suggested by the reviewer, in the introduction we have now underlined that maladaptive ER has not been investigated (lines 151-152). Besides, we also explain that, although people’s repertoire to deal with positive emotions is quite large, we only focused on a limited number of strategies in order to reduce participants’ burden and efforts to complete 3 daily EMA assessments (from line 147).

Finally, it is true that in the discussion we mention three further studies about the influence of positive regulation (more specifically, response modulation) on PE (58, 59, 60). However, these studies are very specific, and they are presented in the discussion in order to explain posteriori why response modulation strategies had a lesser influence on PE in our study. 

This sentence needs more explanation given this study doesn’t study mindfulness, self-esteem, and autonomy: “ Furthermore, intense use of mindfulness has been shown to predict higher levels of daily autonomy (24), whereas positive reappraisal has been associated with greater self-esteem and well-being (25).” Although there is a mindfulness type of item as ER, readers don’t know that until the method, so the sentence appears mostly unrelated.

After stating that “The implementation of positive emotion regulation has been shown to be beneficial for mental health”, our aim was to provide some examples of positive strategies that have been found to be associated with higher well-being, such as counting blessings, capitalizing, mindfulness and positive reappraisal. As the reviewer will see, we have rephrased the paragraph in order to make our objective clearer for the reader (from line 93). We have also deleted the specific terms “self-esteem” and “autonomy”, as these constructs are not explored in our study and might be misleading for the reader. 

A major problem in the paper remains unclear description and treatment of ER strategies. If the individual strategies are being collapsed into categories, why isn’t this done consistently? Some analyses examine ER as individual strategies (all 6) and others include combined strategies (AD, CC, RM) but there is no rationale for the switch back and forth.

We agree with the reviewer that this point was not enough clear in the manuscript. As explained in the previous comment, the rational used in this study was mainly based on the results of the review by Quoidbach et al., (2015), who found attentional deployment, cognitive change and response modulation to be more effective in increasing PE in the brief term. Starting from these 3 categories, we selected two strategies for each category based on the previous literature relating positive emotion regulation to positive emotions. This rational allowed us to explore the association between positive emotion regulation and positive emotions both at a strategy and category levels: In other words, not only the association between the use of 6 positive strategies and PE, but also the influence of strategy category on the experience positive emotional states (i.e., the intrinsic attentional, cognitive or behavioural nature of the strategies). We have now modified the manuscript in order to clarify this rational. As the reviewer will see, we have included a more detailed explanation of the strategies/categories in order to explore the association between positive emotion regulation and PE both at a strategy and category levels (from line 153), as well as a better elaboration of the 3 aims of the study (from line 197). Furthermore, the statistical analyses section has been improved in order to explicitly match each statistical step to the three aims of the study (lines 294-295; lines 309-310; lines 320-321).

The use of acronyms for AD, CC, and RM are confusing as they are not well known. Can these be written out?

We have now substituted acronyms with the entire name of each strategy category. 

It is not clear from Table 1 that authors are correlating scores that define other scores (e.g., expressing and sharing are RM, so that’s why the correlations are .90). This should be explained in a note or authors should pick one (either the aggregate or individual scores) and use one throughout the paper.

Following the reviewer’s suggestion, we have improved the caption of Table 1 in order to include a note about how emotion regulation categories were calculated. We have also modified the table in order to emphasize and separate the two levels (strategy and categories). Furthermore, we have moved the EMA items from the methods to Table 1, in order to make the manuscript easier to read and reduce the use of parentheses, as suggested by the reviewer in the comment below. 

Do authors test for significant differences between strategy use? It is not clear based on how it is written and it’s not clear how much value this text adds.

In the previous version of this manuscript, we did not test for significant differences between the mean use of our strategies of interest. The paragraph pointed by the reviewer was included only to display descriptive statistics. We added this paragraph to be consistent with previous studies on emotion regulation in everyday life, that contained such a paragraph. Nevertheless, as the reviewer suggested, this paragraph does not add enough value to our manuscript to be conserved. More specifically, this paragraph described statistics that are already easily accessible on Table 1. For this reason, rather than testing mean differences which is not very informative and valuable in relation to the aims of our study, we have decided to remove the paragraph.

Why is T1 PE controlled for when predicting ER strategies from PE T0? Authors should have a good rationale for including it and explain it in the paper (or instead not include it as a covariate).

The rational for including PE at t1 as a covariate was already described in the “statistical analyses” section. We have rephrased the sentence in order to make it clearer (lines 302-308). Not controlling for PE t1 could produce a biased estimation of the effect of PE t0 on subsequent strategies. As PE t1 was related to (1) PE t0 and (2) strategy changes from t0 to t1, it could represent a confounding variable when attempting to determine the specific relationship between PE t0 and strategy changes from t0 to t1. Therefore, to ensure that the effect of PE t0 on strategy changes from t0 to t1 was not actually explained by PE t1’s relationships with both variables, we controlled for PE t1.

Conversely, it seems that T0 PE should be included as a covariate when predicting PE T1 from ER. Was that done? Where all 8 strategies run in separate models? 

PE at T0 was included as a covariate when predicting PE at T1 from ER, and the rational for including it was discussed in the “statistical analyses” section. We have underlined this part in the manuscript (lines 312: “PE at t0 was also included as a control variable to neutralize the so-called regression towards the mean effect”). In this case, we only ran one model containing change in the use of each strategy from t0 to t1 as independent variables to predict PE at T1. This information has been clarified in the manuscript (line 310). As the manuscript included a table to report the results of all the analyses performed, we have also included a table to report these results (table 3), which in the previous version of the manuscript were only reported in the text.

Why isn’t AD included in Tables 3 and 4 and corresponding analyses? More information should be included about the models testing these hypotheses, and potentially including analyses in a table given they are major findings of the paper.

The analyses reported in table 4 and table 5 explored the presence of a significant categorical (strategy category: attentional deployment, cognitive change, response modulation) by continuous (PE at t0 for table 4 and change in strategy intensity at t1 for table 5) interaction. A significant interaction means that the slope of the continuous variable is different for one or more levels of the categorical variable. Results are always to be interpreted in relation to the reference group, which in our case was attentional deployment (and that’s why attentional deployment is not reported). We have now modified the tables and included a note in the caption in order to clarify this point (from line 398).

The rationale for the analysis in Table 4 is not provided and the analyses and results are not clear. It is recommended these are removed or set up as a hypotheses and clarified. The authors said they added the analyses in Table 3 and 4 as hypotheses, but the one sentence added at the end of the introduction is not clear in terms of corresponding to Table 4.

Thank you for this suggestion. We have now included more details to describe the third objective of the study (that is, whether strategy category moderates the association between PE and positive emotion regulation) and we have explicitly mentioned that the moderating role of strategy category will be explored in both direction (lines 210-217): Strategy category as a moderator of the association between PE at t0 and positive regulation at t1 (which refers to the analyses reported in table 4); and strategy category as a moderator of the association between positive regulation at t0 and PE at t (which refers to the analyses reported in table 5). Also, we have modified the statistical analyses section in order to explicitly match the analyses conducted to explore the third aim of the study (from line 320). 

The description of some of the results is not summarized as clearly as it could. For example: “Interestingly, the tendency to adopt positive strategies to increase PE was less effective when using RM strategies (e.g., sharing and emotional expression).” Was this difference significant? What was it less effective than?

The difference was statistically significant, as reported in the results section We have now revised this sentence in order to clarify the meaning and interpretation of this result (from line 471).

In general, it is hard to see breaking of paragraphs, but it appears the last paragraph of the paper goes on for two pages. They should break it into two paragraphs (last 3 sentences could be on their own).

The whole manuscript has been revised in order to make paragraphs more concise.

Several typos are present. For example: a. Langstone instead of Langston; b. In the measures paragraphs there are multiple typos, and use of so many parentheses is very hard to follow.

Thank you for pointing out the typos, we have now corrected them. Furthermore, the manuscript has been edited once more by an English-speaking native in order to correct grammatical issues.

---

## [Decision Letter · Decision Letter 2]

29 Apr 2021

Savoring the present: The reciprocal influence between positive emotions and positive emotion regulation in everyday life

PONE-D-20-28135R2

Dear Dr. Colombo,

We’re pleased to inform you that your manuscript has been judged scientifically suitable for publication and will be formally accepted for publication once it meets all outstanding technical requirements.

Kind regards,

Fuschia M. Sirois, PhD

Academic Editor

PLOS ONE

Additional Editor Comments (optional):

Reviewers' comments:

Reviewer's Responses to Questions

**Comments to the Author**

1. If the authors have adequately addressed your comments raised in a previous round of review and you feel that this manuscript is now acceptable for publication, you may indicate that here to bypass the “Comments to the Author” section, enter your conflict of interest statement in the “Confidential to Editor” section, and submit your "Accept" recommendation.

Reviewer #2: All comments have been addressed

2. Is the manuscript technically sound, and do the data support the conclusions?

Reviewer #2: Yes

3. Has the statistical analysis been performed appropriately and rigorously? 

Reviewer #2: Yes

4. Have the authors made all data underlying the findings in their manuscript fully available?

Reviewer #2: Yes

5. Is the manuscript presented in an intelligible fashion and written in standard English?

Reviewer #2: Yes

6. Review Comments to the Author

Reviewer #2: Authors addressed previous concerns and questions. As a result, the manuscript was clearer. Overall, the paper is interesting and it should make a contribution to the literature.

7. PLOS authors have the option to publish the peer review history of their article (what does this mean?). If published, this will include your full peer review and any attached files.

Reviewer #2: No

---

## [Editor Report · Acceptance letter]

3 May 2021

PONE-D-20-28135R2 

Savoring the Present: The Reciprocal Influence Between Positive Emotions and Positive Emotion Regulation in Everyday Life 

Dear Dr. Colombo:

I'm pleased to inform you that your manuscript has been deemed suitable for publication in PLOS ONE. Congratulations! Your manuscript is now with our production department. 

Kind regards, 

on behalf of

Dr. Fuschia M. Sirois 

Academic Editor

PLOS ONE